# Interface-induced dual-pinning mechanism enhances low-frequency electromagnetic wave loss

Bo Cai[1], Lu Zhou[1], Pei-Yan Zhao[1], Hua-Long Peng[1], Zhi-Ling Hou [2], Pengfei Hu [3] ✉, Li-Min Liu [4] ✉ & Guang-Sheng Wang [1] ✉

Improving the absorption of electromagnetic waves at low-frequency bands (2-8 GHz) is crucial for the increasing electromagnetic (EM) pollution brought about by the innovation of the fifth generation (5G) communication technology. However, the poor impedance matching and intrinsic attenuation of material in low-frequency bands hinders the development of low-frequency electromagnetic wave absorbing (EMWA) materials. Here we propose an interface-induced dual-pinning mechanism and establish a magnetoelectric bias interface by constructing bilayer core-shell structures of $NiFe_2O_4$ (NFO) @$BiFeO_3$ (BFO)@polypyrrole (PPy). Such heterogeneous interface could induce distinct magnetic pinning of the magnetic moment in the ferromagnetic NFO and dielectric pinning of the dipole rotation in PPy. The establishment of the dual-pinning effect resulted in optimized impedance and enhanced attenuation at low-frequency bands, leading to better EMWA performance. The minimum reflection loss ($RL_{min}$) at thickness of 4.43 mm reaches -65.30 dB (the optimal absorption efficiency of 99.99997%), and the effective absorption bandwidth (EAB) can almost cover C-band (4.72 ~ 7.04 GHz) with low filling of 15.0 wt.%. This work proposes a mechanism to optimize low-frequency impedance matching with electromagnetic wave (EMW) loss and pave an avenue for the research of high-performance low-frequency absorbers.

The fifth generation (5G) communication technology is developing rapidly with advantages of high network speed, low latency, high reliability, and low-power mass connectivity[1,2]. However, the incidental electromagnetic (EM) pollution is becoming increasingly serious, and the high-performance electromagnetic wave absorbing (EMWA) materials that can effectively absorb EM pollution are urgently needed[3]. Although high-performance EMWA materials can be obtained through material design strategies[4,5], the intrinsic wave impedance, which determines the external electromagnetic wave (EMW) incidence ratio and microwave dissipation capability, inevitably become

mismatching with decreasing frequency, leading to poor EMWA performance at 5G frequency band (2–8 GHz)[6]. Therefore, exploring high-performance low-frequency EMWA strategies has become a top research priority.

Constructing heterogeneous structures is a feasible approach to overcome the intrinsic limits of EMWA materials, of which heterogeneous interface engineering play a very important role in optimizing the impedance matching and EMW attenuation[7–9]. Examples include interfacial polarization[10–12], multiple scattering[13,14], and defect modulation[15,16], and these interfacial interactions thus have

[1]School of Chemistry, Beihang University, Beijing 100191, China. [2]College of Mathematics and Physics & Beijing Key Laboratory of Environmentally Harmful Chemical Analysis, Beijing University of Chemical Technology, Beijing 100029, China. [3]Research Institute of Aero-Engine, Beihang University, Beijing 100191, China. [4]School of Physics, Beihang University, Beijing 100191, China. ✉e-mail: hupengfei@buaa.edu.cn; liminliu@buaa.edu.cn; wanggsh@buaa.edu.cn

fundamental effects on dipole polarization, conduction loss, and magnetic response[17–19], which could probably achieve controllable tuning of EMW absorption. The exchange bias effect between ferromagnetic (FM) and antiferromagnetic (AFM) materials, as a specific interface effect, could be a significant mechanism to adjust the EM parameters of the material[20]. Che et al. constructed an AFM–FM system by forming Ni-NiO heterojunction on the surfaces of NiO nanoplates. This structure induces pinning effect of the AFM phase (NiO) on the FM phase (Ni) and contribute to increasing permeability constant, which is more favorable for impedance matching and magnetic loss[21]. Therefore, the in-depth and extended study of the interfacial pinning effect could provide a significant guidance for rational designing of low-frequency EMWA materials.

In this work, we propose an interface-induced dual-pinning mechanism and designed a distinct magnetoelectric bias interface through constructing bilayer core-shell structures of $NiFe_2O_4$ (NFO) @$BiFeO_3$ (BFO)@polypyrrole (PPy) benefiting from the dual properties of ferromagnetism and magnetostriction of NFO, the dual properties of antiferromagnetism and piezoelectricity of BFO, combined with the strong ability of electron polarization of the conductive polymer PPy[22]. The interfacial magnetic pinning effect of the antiferromagnetic BFO on the ferromagnetic NFO is achieved by the magnetic bias effect, which effectively improves the magnetocrystalline anisotropy of the material, which in turn improves the low-frequency permeability and optimizes the impedance in the low-frequency bands. Meanwhile, the internal bias electric field generated by magnetoelectric driver NFO@BFO plays a role of electric field pinning on the inversion of dipoles, strengthening the relaxation of heterostructure and dielectric loss. The establishment of the dual-pinning effect resulted in optimized impedance and enhanced attenuation at low-frequency bands, leading to better EW performance. Our work here promotes the in-depth comprehension of magnetoelectric bias interface mechanism and provides an approach for the rational design of high-performance low-frequency EW absorbers.

## Results and discussion
### Design and structures
In order to study the mechanism of low-frequency EMW loss induced by heterogeneous interface, a magnetoelectric bias interface was designed and fabricated. Firstly, a core-shell structure of NFO and BFO was constructed to form a magnetic bias interface. Owing to the prominent magnetic pinning[23], the rotation of magnetic moments in the ferromagnetic NFO was stuck by the uncompensated magnetic moments around the interface in the antiferromagnetic BFO. Secondly, based on the piezoelectric properties of the BFO and the magnetostrictive properties of the NFO, an internal bias electric field was established[24], which could cause electric pinning effect to dipolar of conductive polymers such as PPy and further enhance the polarization loss. Thus, NFO@BFO@PPy was rationally designed and synthesized through a simple wet-chemistry method. (Fig. 1a). The octahedra stereoscopic NFO@BFO was wrapped by PPy nanolayers to form core-shell structure (Supplementary Figs. 1–4). The XRD pattern (Fig. 1b) as well as TEM results (Supplementary Figs. 5, 6) demonstrate the coexistence of the cubic spinel (Fd3m) $NiFe_2O_4$ and the rhombohedral perovskite (R3c) $BiFeO_3$, and the existence of PPy was proved by the Fourier transform infrared (FTIR) spectrum (Fig. 1c, Supporting Information)[25,26]. Energy dispersive spectroscopy (EDS) mapping (Fig. 1e) analysis further proved the well-defined core-shell structure. Thus, the bilayer core-shell NFO@BFO@PPy nanocomposites with magnetoelectric bias interface were successfully synthesized.

### Magnetic pinning mechanism
At first, the mechanism of interface-induced magnetic pinning was investigated. According to the Snoek limit (Supporting Information)[27],

the critical limitation for low-frequency impedance matching lies in the difficulty in regulating the natural resonance frequency, which is determined by magnetocrystalline anisotropy field ($H_k$)[28], into GHz range. Therefore, improving the equivalent field of magnetocrystalline anisotropy becomes the key to solve this limitation.

The enhancement of the NFO@BFO for magnetocrystal anisotropy can be illustrated by assuming that the FM magnetic moment of the NFO particles generates an exchange magnetic field $\mu_0 H_{ex}$, acting on the uncompensated magnetic moment of the interface AFM of the BFO shell layer. The corresponding energy difference, $\Delta E = -\frac{1}{2}\Delta\chi_{AF}\mu_0 H_{ex}^2$ (where $\Delta\chi_{AF}$ is the difference in the magnetization rate), could contribute to pin the reversal of the magnetic moment and produce the exchange bias phenomenon[29]. Thus, $\Delta E$ can be considered as an additional magnetic anisotropy term (Fig. 2a).

As a proof of concept, we constructed NFO@BFO heterogeneous structure and the pinning effect between AFM and FM could effectively enhance the magnetocrystalline anisotropy. As proved by the hysteresis loops in Figs. 2b, c, the core-shell structured NFO@BFO sample possesses a hysteresis loop (M–H curves) displaced along the magnetic field axis after field cooling to room temperature treatment compared to the physically co-mingled NFO/BFO sample, whose asymmetry proves the existence of an exchange bias ($H_{EB} = -3.69$ mT) in the NFO@BFO sample at room temperature (Supplementary Fig. 7)[30]. The bifurcation of ZFC and FC curves also revealed the coexistence and the exchange bias phenomenon of FM and AFM phases at room temperature (Supplementary Fig. 8)[31]. The evaluation of the Switching Field Distribution (SFD) curves further proved to a stronger exchange coupling of NFO@BFO (Fig. 2d)[32].

The exchange bias phenomenon originates from the magnetic pinning effect at the interface between the AFM and FM phases, and such pinning effect could lead to increasing $H_k$[33]. Following the Stoner and Wohlfass model, $H_k$ can be expressed after the following equation: $H_k = \frac{2|K_{eff}|}{\mu_0 M_s}$, where $K_{eff}$ is effective magnetic anisotropy constant[34]. $H_k$ could be estimated by comparing the hysteresis lines of the two using the S-W approximation[35]. Its simplified formula[36] is $M = M_s\left(1 - \frac{b}{H^2}\right)$, which the slope $b$ is given by: $b = \frac{8K_{eff}^2}{105\mu_0^2 M_s^2}$, from which it can be concluded that $H_k = c \cdot b^{\frac{1}{2}}$ ($c$ is a constant). It is clear that the slope of the fitted straight-line $b$ is larger for the NFO@BFO ($b = 8.52$) compared to NFO/BFO ($b = 5.68$) (Fig. 2e), which in turn is calculated to have a larger $H_k$. Therefore, NFO@BFO forms a magnetic bias interface through the core-shell structure, which in turn yields a higher $H_k$.

The magnetic permeability of different samples was analyzed to verify the mechanism of enhanced EMW absorption by the magnetic pinning effect. The experimental results (Fig. 2f, g) show that the core-shell structure NFO@BFO@PPy NPs have a higher initial $\mu'$ of 1.10 than the physically co-mingled NFO/BFO@PPy NPs due to increasing $M_s$. More significantly, the $\mu''$ values of NFO@BFO@PPy NPs are higher than for NFO/BFO@PPy NPs in the low-frequency region, while the resonance peak of NFO@BFO@PPy NPs appears at 3.20 GHz. NFO@BFO@PPy NPs have both high $\mu'$ and $\mu''$ i.e., high $\mu_r$, in the low-frequency bands. The above analysis shows the interface-induced magnetic pinning breaks the snoek limit and produces a higher magnetic permeability, leading to optimized impedance matching in low-frequency bands. Meanwhile, the high permeability enhances the magnetic losses, which mainly originate from the resonance losses in the low-frequency bands (Supplementary Fig. 9)[37–39].

### Dielectric pinning mechanism
The attenuation of EW in low-frequency bands is another restrictive factor in addition to satisfying impedance matching. NFO@BFO@PPy has a large magnetic permeability due to the magnetic pinning mechanism, which in turn generates a large magnetic loss, and at the

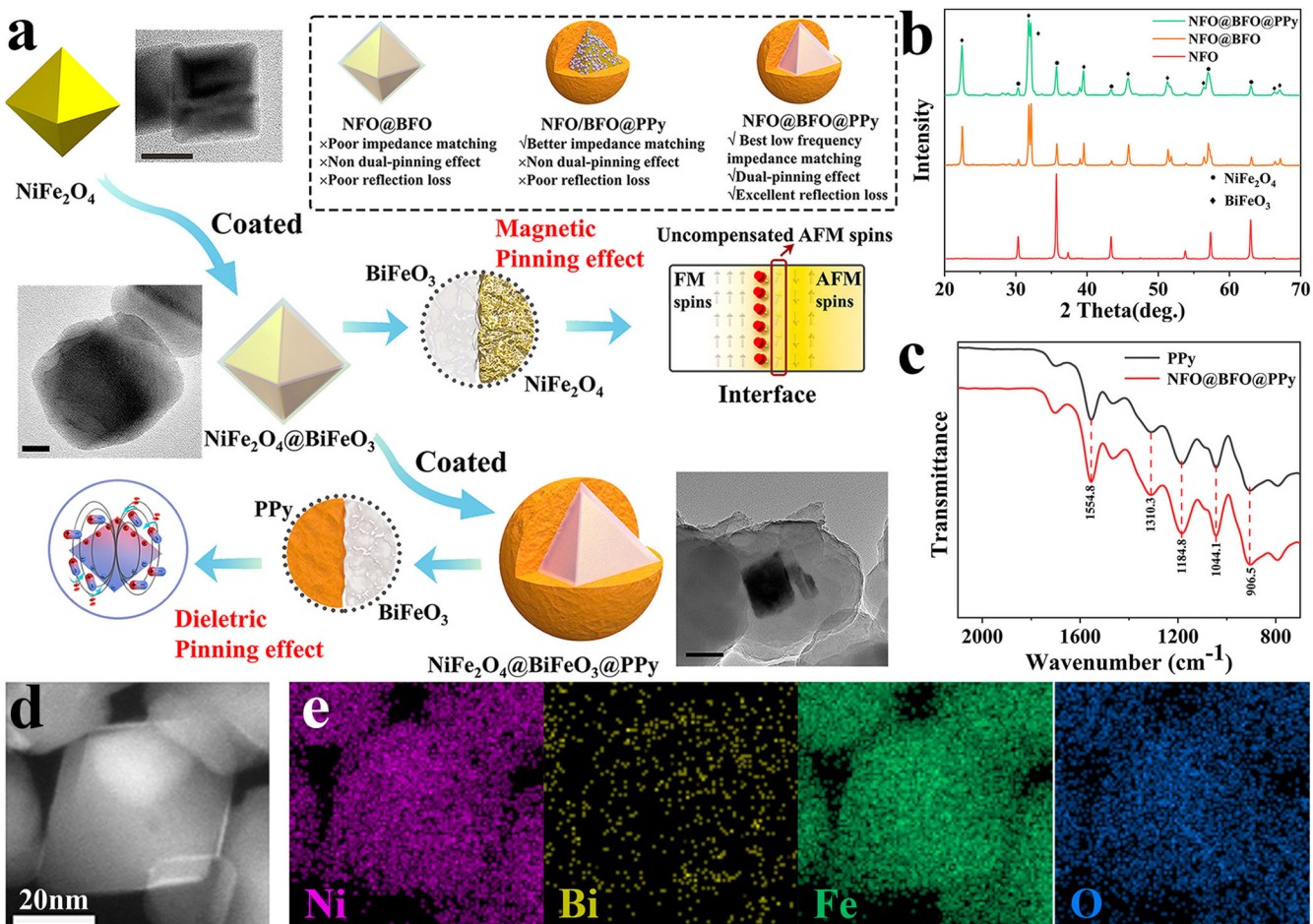

**Fig. 1 | Scheme of the synthesis route and characterization of bilayer core-shell NiFe₂O₄@BiFeO₃@ polypyrrole (NFO@BFO@PPy) nanoparticles (NPs).**
**a** Scheme of the synthesis route of NFO@BFO@PPy NPs and the mechanism of dual-pinning effect. The inset images were obtained by transmission electron microscopy (TEM). Scale bars are as follows: 20 nm (NFO), 20 nm (NFO@BFO), 100 nm (NFO@BFO@PPy). The red box in the interface inset represents the uncompensated antiferromagnetic (AFM) spins formed at the ferromagnetic (FM) and AFM interface. The blue circle in the inset indicates the hindering effect of the surface electric field generated in the BFO on the dipole deflection in the PPy. **b** XRD plot of NFO, NFO@BFO, and NFO@BFO@PPy. **c** FTIR plot of NFO@B-FO@PPy and PPy. **d** STEM of NFO@BFO@PPy. **e** the EDX elemental mapping images of NFO@BFO NPs. Source data are provided as a Source Data file.

same time, its magnetoelectric bias interface could further induce a dielectric pinning effect to increase its polarization loss. More specifically, the exchange bias magnetic field induces the magnetostriction of NFO[40] and subsequently stimulates piezoelectricity of BFO, forming an internal bias electric field ($NFO@BFO+Magnetic\ Fields \to NFO@BFO(e^−+h^+)$)[41]. This surface electric field could pin the repeated rotation of the dipoles in the PPy shell layer under the alternating EM field (Supplementary Fig. 10)[42], further enhancing the relaxation loss caused by polarization in low-frequency bands (Fig. 3a)[43].

The magnetoelectric conversion properties of core-shell NFO@BFO NPs were characterized by piezoelectric response force microscopy (PFM) (Fig. 3b, c). The obtained phase loop demonstrates that the BFO shell exhibits polarization reversibility regardless of the presence of applied magnetic field. In the absence of magnetic field, the coercivity voltages of the BFO shell are −31.97 V and 55.81 V, while the coercivity voltages with the application of magnetic field are −23.99 V and 32.04 V. The smaller coercivity voltages indicate that the strain generated in the magnetostrictive NFO core is effectively transferred to the piezoelectric material BFO shell, which promotes the polarization reversal process in the BFO. Besides, the variation in positive coercivity voltage (22.37 V) is larger than variation in negative coercivity voltage (7.98 V), illustrating the center of the piezoelectric response loop is shifted by the magnetic field, which proves that the magnetoelectric effect generates a bias electric field[44]. The magnetoelectric coupling coefficient is calculated to be $3.70 \times 10^6$ V·cm⁻¹·Oe⁻¹, which have the same order of magnitude as the direct magnetoelectric coupling coefficient values estimated for other core-shell nanostructures using similar measurement methods[41,45]. This demonstrates the strain-mediated magnetoelectric effect renders NFO@BFO as a distinct magnetoelectric coupling actuator. Theoretical simulations further validated the exchange bias field function that induces magnetostriction in the NFO core and simultaneously initiates the BFO shell potential polarization. This proves that the NFO@BFO NP can induce a local surface potential when subjected to a bias magnetic field (Fig. 3d–g and Supplementary Figs. 11, 12)[41].

Benefit from the electric pinning effect, the dielectric permittivity of composites is significantly optimized at low-frequency bands. As shown in Supplementary Fig. 13, the bilayer core-shell structure NFO@BFO@PPy NPs have higher $\varepsilon'$ (≈10.87) and $\varepsilon''$ (≈4.23) compared to with the physically co-hybrid core NFO/BFO@PPy NPs, indicating enhanced energy storage and dissipation capacity of electric energy with dielectric loss capability. The increase of magnetic loss (tan $\delta_\mu$) and dielectric loss (tan $\delta_\varepsilon$) could arise from dual-pinning interface effect, which enhances dipole polarization losses, interface losses, and magnetic losses (Supplementary Fig. 14)[43,46].

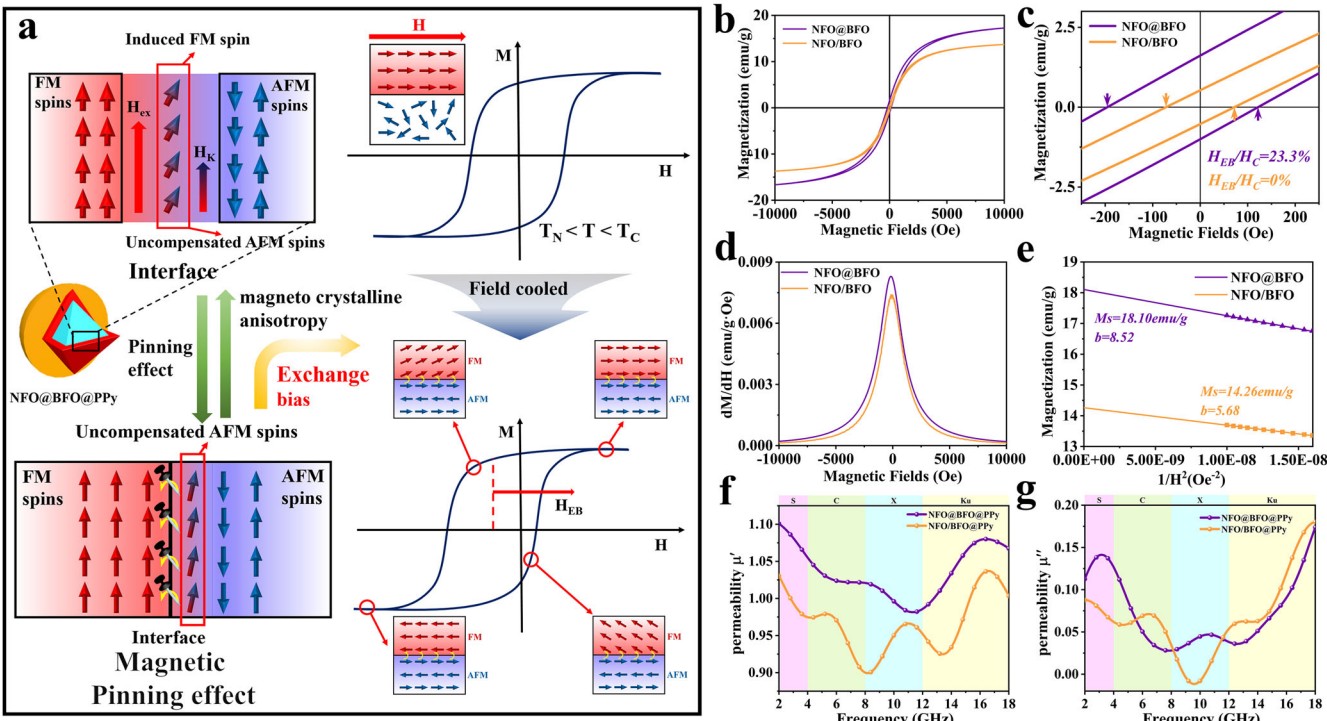

**Fig. 2 | Schematic and experimental proof of the magnetic pinning mechanism. a** Schematic diagram of the magnetic pinning mechanism. The red and blue shades represent the ferromagnetic (FM) and antiferromagnetic (AFM) phases, respectively. Exchange bias is a phenomenon that occurs when ferromagnetism (NFO) and antiferromagnetism (BFO) are coupled to produce a shift in the hysteresis loop through field cooling at temperatures above the Néel temperature ($T_N$) and below the Curie temperature ($T_C$). $H_{EB}$ denotes the exchange bias field. **b** M–H curves. **c** Enlarged M–H curves for NFO@BFO and NFO/BFO. **d** SFD plot. **e** M vs 1/$H^2$ plots. The $R^2$ of the linear fits for NFO@BFO and NFO/BFO are 0.9988 and 0.9989, respectively. Analysis of magnetic parameters of NFO@BFO@PPy and NFO/BFO@PPy **f** permeability real part. **g** permeability imaginary part. Source data are provided as a Source Data file.

## Electromagnetic wave absorbing properties

To verify the interface-induced dual-pining effect, the EM parameters of different samples were calculated and analyzed (Supplementary Figs. 15, 16)[47–49]. The NFO@BFO@PPy sample with bilayer core-shell possess significant enhancement of EMW absorption property, benefiting from the dual-pining mechanism induced by the magnetoelectric bias interface. Specifically, compared with NFO/BFO@PPy samples that do not act by this mechanism, a much lower of −65.30 dB (the optimal absorption efficiency of 99.99997%)[15] can be achieved at 5.68 GHz for NFO@BFO@PPy with a thickness of 4.43 mm. (Fig. 4a, b). The effective absorption bandwidth (EAB) of the NFO@BFO@PPy composite is 2.34 GHz (4.72-7.04 GHz) and shifts toward lower frequencies with a fill rate of only 15.0 wt.% (Supplementary Fig. 17). Two important parameters that determine the $RL_{min}$, i.e., impedance matching ($M_Z$) and attenuation constant ($\alpha_A$), are further analyzed (Supplementary Fig. 18)[50]. On the one hand, NFO@BFO@PPy exhibits a lower frequency impedance matching under the dual-pinning mechanism (Fig. 4c); on the other hand, it achieves a great full-band improvement in attenuation compared with NFO/BFO@PPy (Fig. 4d).

To further evaluate the EW stealth capabilities of the synthesized composites, Radar cross-section (RCS) simulation results of perfect electric conductor (PEC) substrates covered by NFO@BFO@PPy and NFO/BFO@PPy composites are figured out as shown in the three-dimensional RCS intensity image, respectively (Fig. 4e, f and Supplementary Fig. 19). It can be seen that the main flap of the NFO@BFO@PPy coated PEC sheet has the lower reflected signal than that of NFO/BFO@PPy, indicating stronger EMWA abilities. Figure 4g also shows the two-dimensional simulated RCS values in the −90° to 90° detection degree range, with the NFO@BFO@PPy coating exhibiting the smallest RCS signal with a peak of −15.04 dBm². Moreover, the

NFO@BFO@PPy coating reduced the RCS more in all angle ranges, proving its potential for wide-angle stealth applications (Fig. 4h)[51].

Comparing this work with low-frequency EMWA materials studied in recent years (Fig. 5, Table S2)[6,52–57], the bilayer core-shell structure NFO@BFO@PPy composite material proposed in this study exhibit a comprehensive radar stealth capability with great reflection loss and high absorption bandwidth with low filling amount in the low-frequency bands, providing valuable guidance for the research of advanced low-frequency EMWA materials.

In summary, a low-frequency absorption mechanism based on a magnetoelectric bias interface-induced dual-pinning mechanism is proposed in this work and validated by constructing a bilayer core-shell structure NFO@BFO@PPy. The establishment of a dual-pinning mechanism based on the synergistic effect of magneto-electric coupling allows the optimization of low-frequency impedance matching and attenuation enhancement, thus improving the effective loss of low-frequency EM waves. EMWA tests and simulations further validate the superiority of this mechanism from both experiments and theories together, achieving a $RL_{min}$ of −65.30 dB at a thickness of 4.43 mm, and an EAB that almost covers the C-band (4.72-7.04 GHz) with a fill rate of only 15.0 wt.%. This work not only broadens the way for the research of low-frequency EMWA materials, but also can support the improvement of the database of EMWA materials.

## Methods
### The synthesis of NiFe₂O₄ (NFO)@BiFeO₃ (BFO) @polypyrrole (PPy)
To fabricate NiFe₂O₄ (NFO) nanoparticles, 0.187 M Hexadecyl trimethyl ammonium bromide (CTAB), 0.123 M FeCl₃·6H₂O and 0.0613 M

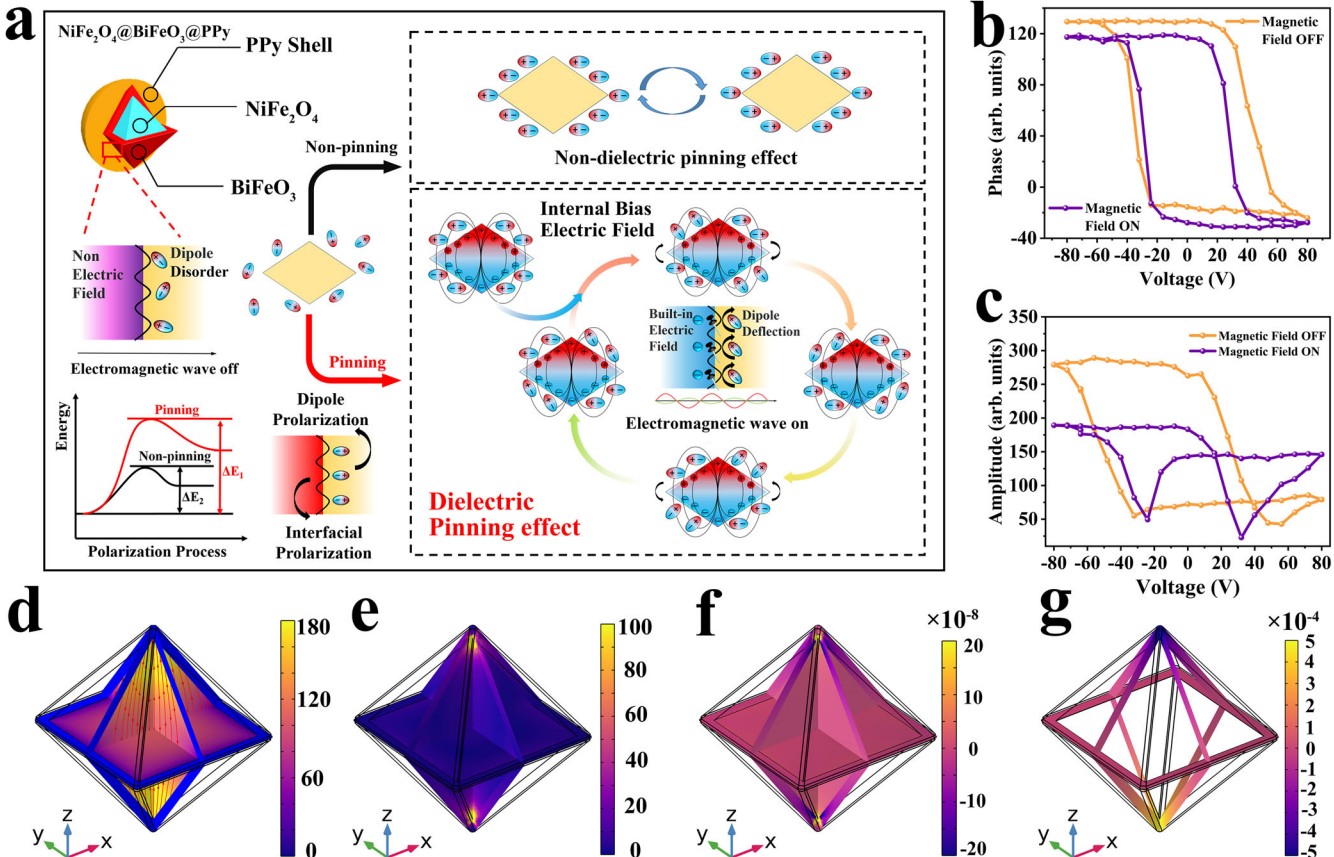

**Fig. 3 | Schematic and experimental proof of the dielectric pinning mechanism.** **a** Schematic diagram of the dielectric pinning mechanism. The yellow shading in the inset indicates that no internal bias electric field is generated, and the red and blue shading represent the positive and negative poles of the internal bias electric field. Benefiting from the core-shell structure of NFO@BFO, the deformation due to magnetostriction of the NFO is transferred to the BFO shell layer, causing it to generate a built-in internal electric field, which acts as a hindrance to the deflection of the dipole in the PPy. **b** Phase response of a single core-shell NP obtained with

and without magnetic field. **c** the corresponding amplitude response. COMSOL simulations of NFO@BFO NPs under a magnetic field of 3.0 mT show **d** the magnetization norm. **e** the strain generated on the BFO shell due to the magnetostrictive effect of NFO core and von Mises stress. **f** the volumetric strain distribution, and **g** the corresponding electric potential induced on the surface of the BFO shell. The legend units are as follows: **d** Oersted (Oe), **e** $10^3$ Pa, **f** $10^{-8}$, and **g** $10^{-4}$ V. Source data are provided as a Source Data file.

NiCl$_2$·6H$_2$O were dissolved in 60 ml of DI water. Then, 20 ml of 6 M NaOH solution was added to the above solution and sonicated under vigorous stirring. Finally, the above solution was transferred to a sealed 100 ml Teflon-lined steel autoclave and heated at 180 °C for 24 h. The obtained powder was washed with DI water and ethanol and dried at 80 °C.

And then, the BiFeO$_3$ (BFO) precursor solution was prepared by dissolving 0.055 M Bi(NO$_3$)$_3$·5H$_2$O and 0.050 M Fe(NO$_3$)$_3$·9H$_2$O in ethylene glycol. The 0.17 g of dried NFO nanoparticles were dispersed into 20 mL of BFO precursor solution using a cell crusher, and then the solution was removed from ethylene glycol in an 80 °C water bath, and the resulting powder was annealed at 600 °C for 2 h at a heating rate of 10 °C·min$^{-1}$. The NFO@BFO nanoparticle was improved by a previously reported method[41].

Finally, 0.100 g of p-toluene sulfonic acid (TsOH), 0.800 g of Polyethylene-polypropylene glycol (P123, average $M_n$-5800), and 9.10 ml of concentrated hydrochloric acid (12 M) were dissolved in 120 ml of DI water. 0.200 g of NFO@BFO nanoparticles were dispersed in the above solution, then 0.400 ml of pyrrole monomer was added and stirred vigorously under an ice water bath. When the temperature of the system reached 0 °C, 80.0 ml of deionized water dissolved with 2.73 g of ammonium persulfate was added and the reaction was carried out for 2 h. The obtained powder was washed with deoxygenated water and ethanol and dried under vacuum at 60 °C to obtain NFO@BFO@PPy nanoparticles.

## Material characterizations

The morphology of the resulting nanoparticles was studied by transmission electron microscopy (TEM, JEOL 2200FS and JEOL 2100F) and scanning electron microscopy (SEM, JSM 7500F). Prior to SEM measurements, the surface of the samples was gold plated using the magnetron sputtering (MSP-1S) apparatus for 1 min 15 s. The distribution of elements along the NFO@BFO nanoparticles was studied using energy dispersive x-ray (EDX) mapping by scanning transmission electron microscopy (STEM, JEOL 2200FS, 200 kV). The local crystal structure was studied by selected area electron diffraction (SAED). The crystal structure of the nanostructures was analyzed by X-ray diffraction (XRD) on a Bruker D8 Advance X-ray diffractometer equipped with a copper target at a wavelength of 1.5418 Å. Fourier transform infrared (FTIR) spectra measurements were performed on FTIR spectrometer (Thermo INX10) instrument.

The magnetic properties were carried out on a Quantum Design superconducting quantum interference device (SQUID) magnetometer (Quantum Design MPMS 3). The temperature dependence of the magnetic moment ($M$–$T$): For the field cooling (FC) process, the sample was cooled from 600 K to 5 K at 200 Oe field. For the zero-field cooling (ZFC) process, the sample was cooled from 600 K to 5 K at zero magnetic field and then heated from 5 K to 600 K at 200 Oe magnetic field. The magnetic hysteresis ($M$–$H$) loop of the sample was measured at 600 K (−1 T-1 T), and the same hysteresis loop test was performed for the sample cooled to 300 K.

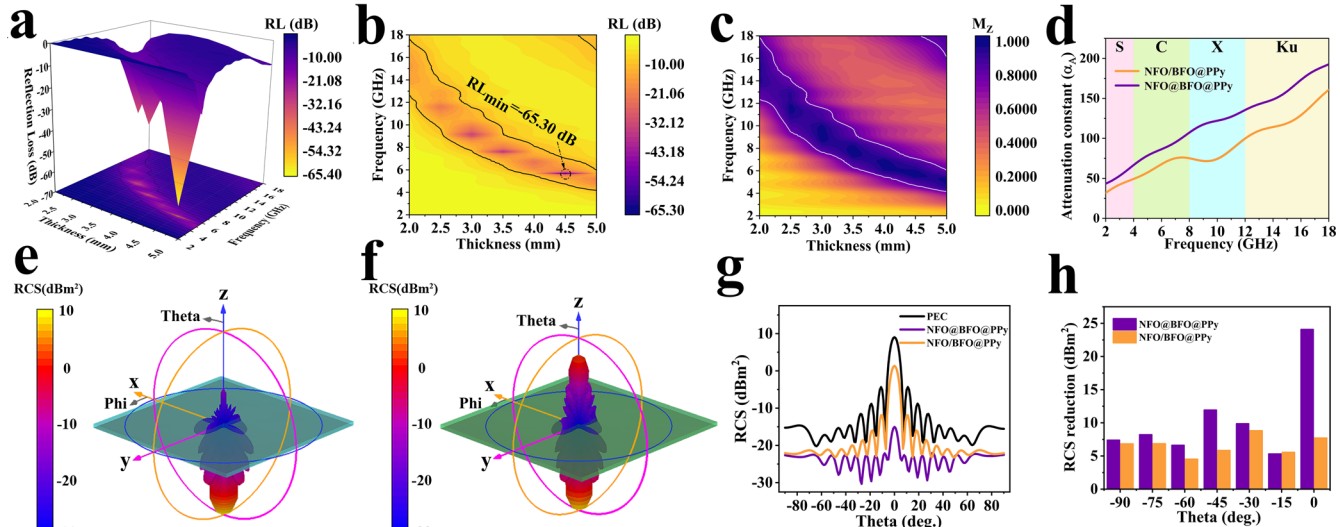

**Fig. 4 | Analysis and comparison of electromagnetic wave absorbing properties of composites.** The reflection loss of NFO@BFO@PPy **a** 3D color mapping surface diagram. **b** 2D color projection. **c** NFO@BFO@PPy impedance matching 2D color projection. **d** comparison of attenuation constants. 3D RCS plot for the PEC substrate covered with **e** NFO@BFO@PPy. **f** NFO/BFO@PPy. **g** RCS simulation curve. **h** RCS reduction at specific angles. Source data are provided as a Source Data file.

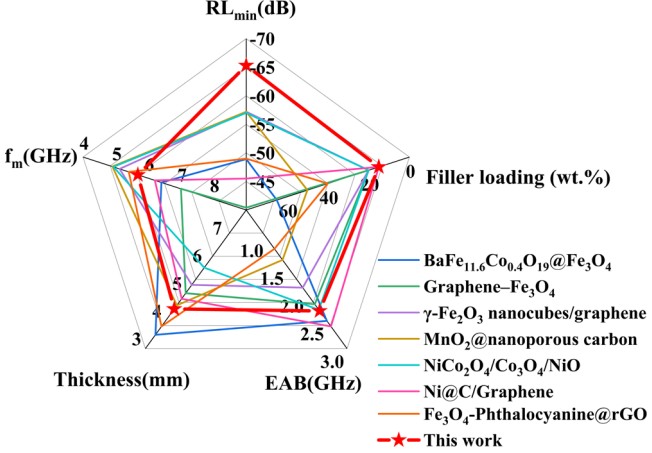

**Fig. 5 | NFO@BFO@PPy composites compared to other low-frequency electromagnetic wave absorbing (EMWA) works.** The $f_m$ is the frequency where $RL_{min}$ is obtained. Source data are provided as a Source Data file.

Piezoelectric reaction force microscopy (PFM) was performed on a commercial atomic force microscope (OXFORD Asylum Research MFP-3D-Bio). Imaging contact force set points were carefully controlled using an ASYELEC-01-R2 probe. To obtain a local piezoelectric response loop, the DC switch is switched from −80 V to 80 V. An in-plane magnetic field of 1000 Oe was applied to the sample in order to study the variation of piezoelectric response under magnetic field.

The EPR characterization was conducted on an electron paramagnetic resonance spectrometer (Bruker A300 10/12, Bruker, Germany).

**Multiphysics simulation of magnetoelectric NFO@BFO NPs**
In this study, we performed simulations in the COMSOL Multiphysics 6.0 software based on similar examples from literature. The magnetoelectric effect simulation of core-shell structure NFO-BFO nanoparticles includes magnetic field, mechanics, and electrostatics. In the simulation, an epitaxially grown BFO shell on the NFO core's [111] plane was considered and implemented accordingly in the model. The thickness of the octahedral NFO NP of the core layer is 94 nm, the thickness of the shell BFO is 20 nm, and the magnetic field intensity is 3.0 mT and applied on the boundaries of the medium along the global z-axis[41].

$$d = \begin{bmatrix} 13.5 & 0 & 0 & 0 & 9 & 0 \\ 0 & 13.5 & 0 & 9 & 0 & 0 \\ 3 & 3 & 50 & 0 & 0 & 0 \end{bmatrix} \cdot 10^{-12}\,[\text{C/N}]$$

$$S_E = \begin{bmatrix} 0.0182167 & -0.0006753 & -0.0179691 & 0.0011707 & 0 & 0 \\ \cdot & 0.0182167 & -0.0179691 & -0.0011707 & 0 & 0 \\ \cdot & \cdot & 0.0492076 & 0 & 0 & 0 \\ \cdot & \cdot & \cdot & 0.0192123 & 0 & 0 \\ \cdot & \cdot & \cdot & \cdot & 0.0192123 & 0.0011707 \\ \cdot & \cdot & \cdot & \cdot & \cdot & 0.0377839 \end{bmatrix}$$
$$\cdot 10^{-9}\,[\text{1/Pa}]$$

**Electromagnetic wave absorption measurement**
The composites used for electromagnetic wave absorption measurement were prepared by mixing the products with PVDF different mass percentages. The mixtures were then pressed into cylindrical-shaped samples ($\Phi_{out}$ = 7.00 mm and $\Phi_{in}$ = 3.04 mm). The complex permittivity and permeability values were measured in the 2–18 GHz range with coaxial wire method by an Agilent N5230C PNA-L Network Analyzer.

**Switching Field Distribution**
The degree of exchange coupling of a sample can be evaluated by the switching field distribution (SFD) curve, the first-order derivative of the demagnetization curve, with a single peak near the zero magnetic field indicating the synchronous magnetic switching process. A stronger peak intensity implies a higher degree of exchange coupling[32].

**The S-W approximation**
The S-W approximation is a method commonly used to determine the magnetocrystalline anisotropy field. It is defined initially as[35]:

$$M = M_s\left(1 - \frac{a}{H} - \frac{b}{H^2}\right) + \chi_p H \tag{1}$$

where $M_s$ is the saturation magnetization and $a$, $b$ and $\chi_p$ are constants. The $a/H$ term is known as the magnetic hardness, $a$ is the constants depends on inhomogeneity of samples that disappears at high magnetic fields. The $\chi_p H$ term is often referred to as the so-called paramagnetism-like term, $X_p$ is the high-field differential susceptibility which is active in the high-temperature analysis. Hence, the $a/H$ and $\chi_p H$ terms can be neglected and the above equation can be reduced to:

$$M = M_s \left(1 - \frac{b}{H^2}\right) \tag{2}$$

The experimental data of magnetization ($M$) versus applied magnetic field ($1/H^2$) becomes linear when $1/H^2$ approaches zero and can be fitted into the positive proportional function above.

## The magnetoelectric coupling coefficient

The magnetoelectric coupling coefficient is defined as[41]:

$$\alpha_{ME} = \frac{\Delta E}{\Delta H} \tag{3}$$

where $\Delta H$ is the amount of magnetic field change and $\Delta E$ is the amount of electric field change per unit length due to the applied magnetic field, respectively. For NFO@BFO NPs, the value of $\Delta E$ can reach:

$$\Delta E = (22.37 - 7.98V)/2/20nm = 3.70 \times 10^9 mV \cdot cm^{-1}$$

Thus, under changing conditions of external magnetic field with $\Delta H$ of 1000 Oe the local magnetoelectric coupling coefficient can be estimated as:

$$\alpha_{ME} = \Delta E/\Delta H = (3.70 \times 10^9 mV \cdot cm^{-1})/1000 Oe$$
$$= 3.70 \times 10^6 mV \cdot cm^{-1} \cdot Oe^{-1}$$

## Reflection loss calculations

Reflection Loss is an important index to evaluate the EMW absorption properties. According to the transmission line theory, the $RL$ value can be calculated by the following formula[47–49]:

$$Z_{in} = \sqrt{\frac{\mu_r}{\varepsilon_r}} \tanh \left| j\left(\frac{2\pi f d}{c}\right) \sqrt{\mu_r \varepsilon_r} \right| \tag{4}$$

$$RL = 20 \log \left| \frac{Z_{in} - 1}{Z_{in} + 1} \right| \tag{5}$$

where $\varepsilon_r$ is the complex permittivity, $\mu_r$ is the complex permeability, $d$ refers to the thickness of the coating, $f$ presents the frequency of the electromagnetic wave, $c$ is the velocity of light in free space ($3 \times 10^8$ m·s$^{-1}$), $Z_{in}$ is the normalized input impedance, respectively.

## Absorption efficiency

The microwave absorption efficiency (%) was calculated from by employing the obtained reflection loss in dB used following equation[15]

$$\text{Absorption efficiency (\%)} = 100 \times \left[1 - 10^{(-|dB/10|)}\right] \tag{6}$$

## Impedance matching calculations

$M_Z$ is used to study the impedance matching characteristics, which can be calculated by the following formula[50]:

$$M_Z = \frac{2Z'_{in}}{|Z_{in}|^2 + 1} \tag{7}$$

Where $Z'_{in}$ means the real normalized input impedance, when $M_Z = 1$, the ideal impedance matching will be realized, indicating that the EMWA performance at this time will be the best.

## The attenuation constant ($\alpha_A$) calculations

The attenuation constant estimates the combined loss capacity of incident electromagnetic waves and represents the EWMA attenuation characteristics of the material, which is determined by the following formula[50]:

$$\alpha_A = \frac{\sqrt{2}}{c} \pi f \sqrt{(\mu''\varepsilon'' - \mu'\varepsilon') + \sqrt{(\mu''\varepsilon'' - \mu'\varepsilon')^2 + (\mu'\varepsilon'' + \mu''\varepsilon')^2}} \tag{8}$$

## Radar scattering cross-section (RCS) calculations

RCS is the most critical concept in radar stealth technology, which characterizes a physical quantity of the intensity of the echoes generated by the target when exposed to radar waves. For a scattering source, its RCS value ($\sigma$) can be expressed by the following equation[6]:

$$\sigma(dBm^2) = 10 \log \left[\frac{4\pi S}{\lambda^2} \left|\frac{E_S}{E_i}\right|^2\right] \tag{9}$$

where $S$ is the area of the plate, $\lambda$ is the length of the incident electromagnetic wave, and $E_S$ and $E_i$ are the electric field strengths of the emitted and received waves, correspondingly. CST STUDIO SUITE 2014 was applied to simulate the RCS values of as-prepared NFO@BFO@PPy and NFO/BFO@PPy composites under open boundary conditions. The simulation model consisted of the perfect electric conductor (PEC) layer at the bottom and an absorbing layer with a thickness of 4.43 mm on the top. The dimension of length was equal to the width of 200 mm. Then, the created model was placed on the $xOy$ plane, and the linear polarized plane electromagnetic wave was added with the incidence direction on $Z$-axis positive to negative, and the electric polarization was along the $X$-axis. In addition, the far-field monitor frequency was set as 5.68 GHz.

## Reporting summary

Further information on research design is available in the Nature Portfolio Reporting Summary linked to this article.

## Data availability

The data supporting the findings of this study are available within the article and the Supplementary Information. Source data are provided with this paper.

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

## Acknowledgements

This work was supported by the National Natural Science Foundation of China (No. 52073010 and 52373259) (G.-S.W.), (No. 52225308, 1197403) (L.-M.L.) and (No. 52371147) (P.H.).

## Author contributions

G.-S.W., P.H. and B.C. conceived this project. B.C., P.-Y.Z. and H.-L.P. prepared the samples. P.H. and B.C. performed TEM observation and analyses. B.C. and Z.-L.H. carried out electromagnetic wave absorption performance tests and mechanistic studies. L.Z. and Z.-L.H. contributed to analog simulation. All authors participated in the discussion. L.-M.L., P.H. and B.C. analyzed the data and cowrote this paper.

## Competing interests

The authors declare no competing interests.
