## [Peer Review File · Nature Communications]

REVIEWER COMMENTS

Reviewer #1 (Remarks to the Author):

The formation of heterogeneous interfaces brings about a series of striking physicochemical properties, such as pinning effects, which have fundamental effects on dipole polarization and magnetic response, thereby improving electromagnetic wave absorption performance. It is foreseeable that heterogeneous interface engineering design will provide an effective and powerful platform for the development of next-generation electromagnetic absorption systems in the coming years. However, understanding and enhancing these interface effects at a deeper level still faces some serious challenges. This work has stimulated the infinite vitality of heterogeneous interface modulation and customization in combinatorial optimization and structural design, explained the heterogeneous interface mechanism from a deeper level, and promoted the development of electromagnetic wave absorption materials towards high performance and intelligent low frequency response. In this manuscript, for the current interest in developing methods for absorption of low-frequency electromagnetic waves, the two-spike core-shell method proposed by the authors is certainly an interesting mechanism to achieve this goal. Thus, I recommend publication of this manuscript after minor revision. Specific comments are listed as follows:

1. The author mentions in the article "the critical limitation for low-frequency impedance matching lies in the difficulty in regulating the natural resonance frequency, which is determined by magnetocrystalline anisotropy field, into GHz range." The authors should illustrate the relationship between the natural resonance frequency and the magnetic permeability.
2. The author claims that "this surface electric field can fix the repetitive rotation of dipoles in the PPy shell under alternating electromagnetic fields, and further enhance the relaxation loss caused by low frequency polarization". Additional experimental or theoretical analysis would be helpful.
3. The electromagnetic wave absorption performance of NFO and BFO should be discussed for comparison.
4. Ref. 5 previously reported the important pinning effect of wave absorption. This work of dual-pinning effect is a meaningful theoretical extension, which could provide guidance for the rational design of microwave absorption materials, especially for low-frequency band. The significance of pinning effect needs to be further clarified.
5. Some secondary structures can be seen on the surface of particles in Supplementary Fig. 1.C, but how these secondary structures are formed and whether they have any influence on interface pinning effect and wave absorption performance is not explained.

Reviewer #2 (Remarks to the Author):

In this manuscript, Interface-induced Dual-pinning Mechanism Enhances Low-Frequency Electromagnetic Wave Loss were studied. However, I have some concerns regarding the said research for the dual pinning mechanism used for the enhancement of low-frequency EM wave losses. Therefore, I think the authors must address the following major comments before the final publication of the article. Here are my comments:

1. In the abstract, the author primarily discusses the importance of pollutant removal used in the manuscript. It is recommended that the authors align the abstract more closely with the manuscript's title, emphasizing the correlation between the research findings and the title.
2. Here we propose an interface-induced dual-pinning mechanism and establish a novel a magnetoelectric bias interface by constructing bilayer core-shell structures, Correct the sentence.
3. In the introduction part, the authors must include the most important relevant articles as; <https://doi.org/10.1016/j.surfin.2023.103377>, <https://doi.org/10.1016/j.ceramint.2022.10.326> <https://doi.org/10.1016/j.jmmm.2022.169955>, <https://doi.org/10.1016/j.ceramint.2022.01.044> <https://doi.org/10.1016/j.ceramint.2021.12.099>
4. The establishment of the dual-pinning effect resulted in optimized impedance and enhanced attenuation at low-frequency bands, leading to better EMWA performance. The author must discuss the performance in terms of percentage.
5. Authors must include the Rietveld refinement of the XRD patterns and include the detailed structural analysis as supplementary material.

6. In the whole manuscript, the authors must revise the language thoroughly. The grammatical errors must be removed.
7. The authors must use C-band instead of c-band in the whole paper.
8. Why did the author choose the NiFe₂O₄ (NFO)@BiFeO₃ (BFO)@polypyrrole (PPy), provides some significance in the manuscript.
9. The authors must provide a detail discussion about the establishment of the dual-pinning effect resulting in optimized impedance and enhanced attenuation at low-frequency bands, leading to better EW performance.
10. Figure 2 shows the schematic diagram of the magnetic pinning mechanism. b, M-H curves. c, Enlarged M-H curves for NFO@BFO and NFO/BFO. d, SFD plot. e, M vs 1/H₂ plots. Analysis of magnetic parameters of NFO@BFO@PPy and NFO/BFO@PPy f, permeability real part. g, permeability imaginary part. Authors must explain why they did not do the magnetic loops analysis of NFO@BFO@PPy and NFO/BFO@PPy composites.
11. Why NFO@BFO@PPy have a large magnetic permeability? Explain with more details.
12. The smaller coercivity voltages indicate that the strain generated in the magnetostrictive NFO core is effectively transferred to the piezoelectric material BFO shell, which promotes the polarization reversal process in the BFO.? Explain and justify.
13. a much lower RL_{min} of -65.30 dB can be achieved at 5.68 GHz for NFO@BFO@PPy with a thickness of 4.43 mm. (Fig. 4,a,b). The effective absorption bandwidth (EAB) of the NFO@BFO@PPy composite is 2.34 GHz (4.72 ~ 7.04 GHz) and shifts toward lower frequencies with a fill rate of only 15.0% (Supplementary Fig. 15). Why the thickness was too large. Explain and justify
14. Why the RCS reduction is too high in NFO@BFO@PPy?
15. Why ethylene glycol and p-toluenesulfonic acid was used? Explain its importance.
16. Which version of COMSOL Multiphysics software is used. Mention in the text.
17. CST STUDIO SUITE was applied to simulate the RCS values of as-prepared NFO@BFO@PPy and NFO/BFO@PPy composites under open boundary conditions. Mention the version in the text.
18. Conclusion must be brief

Response to Reviewers' Comments

Reviewer 1:

Comments: *The formation of heterogeneous interfaces brings about a series of striking physicochemical properties, such as pinning effects, which have fundamental effects on dipole polarization and magnetic response, thereby improving electromagnetic wave absorption performance. It is foreseeable that heterogeneous interface engineering design will provide an effective and powerful platform for the development of next-generation electromagnetic absorption systems in the coming years. However, understanding and enhancing these interface effects at a deeper level still faces some serious challenges. This work has stimulated the infinite vitality of heterogeneous interface modulation and customization in combinatorial optimization and structural design, explained the heterogeneous interface mechanism from a deeper level, and promoted the development of electromagnetic wave absorption materials towards high performance and intelligent low frequency response. In this manuscript, for the current interest in developing methods for absorption of low-frequency electromagnetic waves, the two-spike core-shell method proposed by the authors is certainly an interesting mechanism to achieve this goal. Thus, I recommend publication of this manuscript after minor revision.*

Reply: We are very grateful for the reviewers' insightful review and constructive comments, and welcome the opportunity to address these comments and describe our revisions below.

Specific comments are listed as follows:

Q1: *The author mentions in the article "the critical limitation for low-frequency impedance matching lies in the difficulty in regulating the natural resonance frequency, which is determined by magnetocrystalline anisotropy field, into GHz range." The authors should illustrate the relationship between the natural resonance frequency and the magnetic permeability.*

Reply: Natural resonance can be considered as a special form of ferromagnetic resonance, located in the frequency interval between 0.1 GHz and 10 GHz, where an extreme value of the complex permeability occurs. The natural resonance frequency corresponds to the point at which the imaginary part of the permeability reaches maximum (*J. Magn. Magn. Mater.*, 2004, **281**, 195-205). For conventional spinel ferrite, the intrinsic resonance frequency is usually close to or below 1 GHz (*J. Magn. Magn. Mater.*, 2000, **215-216**, 253-259), and the natural resonance frequency is determined by the magnetocrystalline anisotropy field (*J. Magn. Magn. Mater.*, 2023, **586**, 171159). Therefore, the natural resonance frequency can be expanded to the GHz range by increasing the magnetocrystalline anisotropy field of the magnetic component to obtain a larger permeability, which facilitate achieving impedance matching and enhance attenuation at low-frequency bands (*J. Phys. Appl. Phys.*, 2009, **42**, 155004).

Q2: *The author claims that "this surface electric field can fix the repetitive rotation of dipoles in the PPy shell under alternating electromagnetic fields, and further enhance the relaxation loss caused by low frequency polarization". Additional experimental or theoretical analysis would be helpful.*

Reply: The EPR spectrum of NFO@BFO@PPy reveals that the electrons in PPy shell layer electrons are more delocalized and more easily polarized (*Chem. Eng. J.*, 2022, **430**, 132747). When subjected to an applied electromagnetic field, the bound charge undergoes localized motion and the dipole moment rotates (*Adv. Opt. Mater.*, 2022, **10**, 2200249). With increasing operating frequency, the rotation of the dipole moments cannot keep up with the change of the applied frequency, leading to polarization relaxation (*Carbon*, 2020, **168**, 606-623). During process of polarization relaxation, the repeated deflection of the dipole moment induces loss to the electromagnetic wave (*J. Alloys Compd.*, 2017, **728**, 1065-1075). In the present work, the distinct dielectric pinning effect could effectively prohibit the repeated rotation of the dipole moments, which could not keep pace with relatively low frequency, leading to low-frequency electromagnetic wave loss.

Q3: The electromagnetic wave absorption performance of NFO and BFO should be discussed for comparison.

Reply: We have added the electromagnetic wave absorption performance of NFO@PPy and BFO@PPy as requested by the reviewers.

Fig. R1 The reflection loss of NFO@PPy (a) and BFO@PPy (b).

The RL_{min} of -37.77 dB for NFO@PPy composite is achieved at 10.96 GHz with the thickness of 2.50 mm. BFO@PPy composite reaches a RL_{min} of -43.84 dB at 8.48 GHz and a corresponding thickness of 3.50 mm. From the results of the absorption of electromagnetic waves by a single core component, neither possesses an effective loss of low-frequency electromagnetic waves.

To address this point, we have revised the supplementary information to add details of the experimental data.

Page 20 in the revised supporting information:

Supplementary Fig. 16 The reflection loss of NFO@PPy (a) and BFO@PPy (b).

The RL_{min} of -37.77 dB (the optimal absorption efficiency of 99.98%) for NFO@PPy composite is achieved at 10.96 GHz with the thickness of 2.50 mm. BFO@PPy composite reaches a RL_{min} of -43.84 dB (the optimal absorption efficiency of 99.996%) at 8.48 GHz and a corresponding thickness of 3.50 mm. From the results of the absorption of electromagnetic waves by a single core component, neither possesses an effective loss of low-frequency electromagnetic waves.

Q4: Ref. 5 previously reported the important pinning effect of wave absorption. This work of dual-pinning effect is a meaningful theoretical extension, which could provide guidance for the rational design of microwave absorption materials, especially for low-frequency band. The significance of pinning effect needs to be further clarified.

Reply: Heterogeneous interface engineering plays a crucial important role in optimizing the impedance matching and enhancing EMW attenuation. Che et al. constructed an AFM-FM system by forming Ni-NiO heterojunction on the surfaces of NiO nanoplates. This structure induces pinning effect of the AFM phase (NiO) on the FM phase (Ni). The interfacial pinning effect between ferromagnetic (FM) and antiferromagnetic (AFM) materials, as a special interfacial effect, can modulate the magnetic properties of composites, which may be an important mechanism for tuning the electromagnetic parameters of the materials (*ACS Appl. Mater. Interfaces*, 2018, **10**, 15104–15111). Our work receives inspiration from that work to extend the interface pinning effect, arguing that interfaces can realize not only magnetic pinning at AFM and FM interfaces but also dielectric pinning of the interface electric field to the electric dipole, which is of great significance for the design of rational and adjustable electromagnetic wave absorbing materials.

To address this comment, we have supplemented the presentation of the abstract in the revised manuscript accordingly.

Page 2, Paragraph 2, Line 3-14 in the revised manuscript:

“Examples include interfacial polarization¹⁰⁻¹², multiple scattering¹³⁻¹⁵, and defect modulation^{16,17}, and these interfacial interactions thus have fundamental effects on dipole polarization, conduction loss, and magnetic response¹⁸⁻²⁰, which could probably achieve controllable tuning of EMW absorption. The exchange bias effect between

ferromagnetic (FM) and antiferromagnetic (AFM) materials, as a specific interface effect, could be a significant mechanism to adjust the EM parameters of the material²¹. Che et al. constructed an AFM-FM system by forming Ni-NiO heterojunction on the surfaces of NiO nanoplates. This structure induces pinning effect of the AFM phase (NiO) on the FM phase (Ni) and contribute to increasing permeability constant, which is more favorable for impedance matching and magnetic loss²². Therefore, the in-depth and extended study of the interfacial pinning effect could provide a significant guidance for rational designing of low-frequency EMWA materials.”

Q5: *Some secondary structures can be seen on the surface of particles in Supplementary Fig. 1.C, but how these secondary structures are formed and whether they have any influence on interface pinning effect and wave absorption performance is not explained.*

Reply: The secondary structure on the surface of the particles is due to the magnetron sputtering of the gold particles on the surface of samples before SEM measurement. It should be noted that such secondary structures are not generated during the synthesis of the samples and have no influence on interface pinning effect and wave absorption performance. Considering the comments given by the reviewers, we reduced the time of magnetron sputtering treatment and re-performed the SEM measurements. The new measurements as shown in Fig. 2R indicate that reducing the magnetron sputtering treatment time resulted in a reduction of the surface secondary structure.

Fig. R2 SEM (a,c) and TEM (b,d) images of NiFe₂O₄.

To address this point, we have revised Supplementary Fig. 1 and added experimental details in revised manuscript accordingly.

Supplementary Fig. 1 SEM (a,c) and TEM (b,d) images of NiFe₂O₄.

Page 11, Paragraph 4, Line 3-4 in the revised manuscript:

“Prior to SEM measurements, the surface of the samples was gold plated using the magnetron sputtering (MSP-1S) apparatus for 1min 15s.”

Reviewer 2:

Comments: *In this manuscript, Interface-induced Dual-pinning Mechanism Enhances Low-Frequency Electromagnetic Wave Loss were studied. However, I have some concerns regarding the said research for the dual pinning mechanism used for the enhancement of low-frequency EM wave losses. Therefore, I think the authors must address the following major comments before the final publication of the article. Here are my comments:*

Reply: We are very grateful for the reviewers' recommendation and positive comments. The concerns of the reviewer are thoroughly addressed in a point-by-point manner as follows, upon which the manuscript is revised accordingly.

Here are my comments:

Q1: *In the abstract, the author primarily discusses the importance of pollutant removal used in the manuscript. It is recommended that the authors align the abstract more closely with the manuscript's title, emphasizing the correlation between the research findings and the title.*

Reply: Thanks for the reviewer's constructive suggestion. We revised the abstract as follows:

Page 1, Paragraph 1, Line 1-3 in the revised manuscript:

“Improving the absorption of electromagnetic waves at low-frequency bands (2-8 GHz) is crucial for the increasing electromagnetic (EM) pollution brought about by the innovation of the fifth generation (5G) communication technology.”

Q2: *Here we propose an interface-induced dual-pinning mechanism and establish a novel a magnetoelectric bias interface by constructing bilayer core-shell structures, Correct the sentence.*

Reply: We have corrected the sentence:

Page 1, Paragraph 1, Line 7-9 in the revised manuscript:

“Here we propose an interface-induced dual-pinning mechanism and establish a novel magnetoelectric bias interface by constructing bilayer core-shell structures of NiFe₂O₄ (NFO)@BiFeO₃ (BFO)@polypyrrole (PPy).”

Q3: *In the introduction part, the authors must include the most important relevant articles as;*

<https://doi.org/10.1016/j.surfin.2023.103377>,

<https://doi.org/10.1016/j.ceramint.2022.10.326>

<https://doi.org/10.1016/j.jmmm.2022.169955>,

<https://doi.org/10.1016/j.ceramint.2022.01.044>

<https://doi.org/10.1016/j.ceramint.2021.12.099>.

Reply: We have cited these references as detailed in reference 2, 9, 23, 39, 40.

“2. Zhang, Y., Dai, F., Mouldi, A., Bouallegue, B. & Akhtar, M. N. Tunable microwave absorption features in bi-layer absorber based on mesoporous CuS micro-particle with 3D hierarchical structure and nanosphere like NiCo₂O₄. *Ceram. Int.* **48**, 9146–9156 (2022).”

9. Rajhi, A. A., Alamri, S., Logesh, K., Mohanavel, V. & Akhtar, M. N. Synergistic effect of tuning nanocomposite morphology, composition and layer arrangement for boosting microwave dissipation performance. *J. Magn. Magn. Mater.* **563**, 169955 (2022).
23. Bai, J., Abdelbasset, W. K., Elkholi, S. M., Ismail, K. A. & Akhtar, M. N. Efficient single and bi-layer absorbers of CaTiO₃ micro-cubes and polypyrrole nanotubes composites for enhanced microwave absorption in X and Ku band. *Ceram. Int.* **48**, 11953–11961 (2022).
39. Akhtar, M. N. et al. Microwave absorption, physicochemical, elemental mapping, and high-frequency perspectives of the Co, Cu, Zn doped Ni-Ce absorbers for Ku band frequency. *Surf. Interfaces* **42**, 103377 (2023).
40. Ji, M., Ji, P, F., Mehrez, S., Alamri, S. & Akhtar, M. N. Multi-interface-induced by regulating nanocomposite morphology and absorber design to achieve wideband electromagnetic wave absorber. *Ceram. Int.* **49**, 8071–8080 (2023).”

Q4: *The establishment of the dual-pinning effect resulted in optimized impedance and enhanced attenuation at low-frequency bands, leading to better EMWA performance. The author must discuss the performance in terms of percentage.*

Reply: The microwave absorption efficiency (%) was calculated from by employing the obtained reflection loss in dB used following equation (*Adv. Mater.*, 2023, **35**, 2210243):

Specifically, compared with NFO/BFO@PPy (the optimal absorption efficiency of 99.8%) samples without dual-pinning effect, a much lower RL_{min} of -65.30 dB (the optimal absorption efficiency of 99.99997%) can be achieved at 5.68 GHz for NFO@BFO@PPy with a thickness of 4.43 mm. The effective absorption bandwidth (EAB) of the NFO@BFO@PPy composite is 2.34 GHz (4.72 ~ 7.04 GHz) and shifts toward lower frequencies with a fill rate of only 15.0%.

The reflection loss of the various samples mentioned in the article have been added in their percentage representation.

We have supplemented the corresponding notes in the revised manuscript and revised supplementary material accordingly.

Page 15, Paragraph 2, Line 1-2 in the revised manuscript:

“**Absorption efficiency.** The microwave absorption efficiency (%) was calculated from by employing the obtained reflection loss in dB used following equation¹⁶

$$(6)”$$

Page 1, Paragraph 1, Line 12-14 in the revised manuscript:

“The minimum reflection loss (RL_{min}) at thickness of 4.43 mm reaches -65.30 dB (the optimal absorption efficiency of 99.99997%).”

Page 19, Paragraph 1, Line 3 in the revised supporting information:

“NFO/BFO@PPy composite reaches a RL_{min} of -27.72 dB (the optimal absorption efficiency of 99.831%) at 6.48 GHz and a corresponding thickness of 4.86 mm.”

Page 20, Paragraph 1, Line 1, 3 in the revised supporting information:

“The RL_{min} of -37.77 dB (the optimal absorption efficiency of 99.98%) for NFO@PPy composite is achieved at 10.96 GHz with the thickness of 2.50 mm. BFO@PPy composite reaches a RL_{min} of -43.84 dB (the optimal absorption efficiency of 99.996%) at 8.48 GHz and a corresponding thickness of 3.50 mm. From the results of the absorption of electromagnetic waves by a single core component, neither possesses an effective loss of low-frequency electromagnetic waves.”

Page 21, Paragraph 1, Line 4-6 in the revised supporting information:

“The NFO@BFO@PPy with a fill rate of 15.0% exhibits the RL_{min} of -65.30 dB at 5.68 GHz, corresponding to a thickness of 4.43 mm. The impedance mismatching is caused by the larger fill rate increasing the dielectric constant of the composite, thus, the RL_{min} of NFO@BFO@PPy with a fill rate of 20.0% at 6.88 GHz is -32.27 dB (the optimal absorption efficiency of 99.94%) and the matching thickness is 3.45 mm, while NFO@BFO@PPy with a fill rate of 25.0% reaches a RL_{min} of -17.01 dB (the optimal absorption efficiency of 98.00%) at 7.20 GHz and a corresponding thickness of 2.89 mm.”

Q5: Authors must include the Rietveld refinement of the XRD patterns and include the detailed structural analysis as supplementary material.

Reply: Crystallographic parameters obtained by using the Rietveld refinement and the fraction of the refinement are shown in Table R1, which is able to verify a similar proportion of components in physically co-mingled NFO/BFO and NFO@BFO composites.

Fig. R3 Rietveld refinement results for a) NFO@BFO, b) NFO/BFO.

	A(Å)		Calculated density(g/cm ³)		Fraction(%)	
	NFO	BFO	NFO	BFO	NFO	BFO
NFO@BFO	8.34	5.58(a)/13.87(c)	5.75	8.59	34.7	65.3
NFO/BFO	8.34	5.58(a)/13.87(c)	5.50	8.73	40.1	59.9

Table R1 Crystallographic parameters obtained by using Rietveld refinement.

Page 7 in the revised supporting information:

Supplementary Fig. 6 Rietveld refinement results for NFO NPs (a) and NFO@BFO NPs (b).

Page 8 in the revised supporting information:

	A(Å)		Calculated density(g/cm ³)		Fraction(%)	
	NFO	BFO	NFO	BFO	NFO	BFO
NFO@BFO	8.34	5.58(a)/13.87(c)	5.75	8.59	34.7	65.3
NFO/BFO	8.34	5.58(a)/13.87(c)	5.50	8.73	40.1	59.9

Table S1 Crystallographic parameters obtained by using Rietveld refinement.

Page 4, Paragraph 1, Line 14-17 in the revised manuscript:

“Crystallographic parameters obtained by using the Rietveld refinement and the fractio

of the refinement are shown in Supplementary Fig. 6, Table S1, which is able to verify a similar proportion of components in physically co-mingled NFO/BFO and NFO@BFO composites.”

Q6: *In the whole manuscript, the authors must revise the language thoroughly. The grammatical errors must be removed.*

Reply: We have scrutinized the whole manuscript and carefully corrected the grammatical errors.

Q7: *The authors must use C-band instead of c-band in the whole paper.*

Reply: We have changed c-band to C-band throughout the text.

Q8: *Why did the author choose the NiFe₂O₄ (NFO)@BiFeO₃ (BFO)@polypyrrole (PPy), provides some significance in the manuscript.*

Reply: The interface-induced dual-pinning mechanism comprises two significant components. On the one hand, it is the magnetic pinning effect generated by antiferromagnetic phase and ferromagnetic phase. On the other hand, the magnetoelectric actuator generated by the magnetostrictive composite piezoelectric material produces the dielectric pinning effect of the dipoles generated by the internal bias electric field under the action of magnetic field. The dual synergistic effect contributes to excellent absorption of low-frequency electromagnetic waves.

The heterogeneous structure was rationally fabricated in order to construct the magnetoelectric bias interface, NiFe₂O₄ (NFO) is a magnetostrictive material with ferromagnetic properties and strong magnetic loss (*Chem. Eng. J.*, 2022, **428**, 131160). BiFeO₃ (BFO) is a typical piezoelectric material with room temperature antiferromagnetism, whose electrical, magnetic and structural order temperature is much higher than room temperature (*Nat. Mater.*, 2019, **18**, 580–587). The core-shell structure formed by ferromagnetic NFO and antiferromagnetic BFO generates interfaces with magnetic pinning effect, which in turn constitutes an exchange bias field. At the same time, the magnetoelectric driver formed by the magnetostrictive NFO and the piezoelectric BFO can generate an electric field in the BFO shell under the exchange bias field, which further creates the conditions for the construction of an interface with electric pinning. Polypyrrole (PPy) as a conductive polymer, and has a strong ability of electron polarization, has been widely used in electromagnetic wave absorption with its excellent dielectric loss capability and mild low-temperature preparation process (*Adv. Eng. Mater.*, 2022, **24**, 2100790). The conductive polymer polypyrrole is utilized for coating, which is rich in dipoles, and the generation of internal electric field plays a role of electric field pinning on the inversion of dipoles, strengthening the relaxation of heterostructure and dielectric loss. By combining these materials within the heterogeneous structure, the magnetoelectric bias interface is effectively constructed. Leveraging the interface-induced dual-pinning mechanism, the magnetic pinning effect

enhances the imaginary part of the permeability of the material at low frequency, so that it has more adjustable impedance matching at low frequency. Meanwhile, the dielectric pinning improves the loss of low frequency medium and realizes strong absorption at low frequency. The composite material makes full use of the intrinsic characteristics of the material and the interface effect, is expected to become an excellent low-frequency electromagnetic wave absorption material.

To address this comment, corresponding description was added in the revised version.

Page 2, Paragraph 3, Line 3-6 in the revised manuscript:

In this work, we proposed an interface-induced dual-pinning mechanism and designed a distinct magnetoelectric bias interface through constructing bilayer core-shell structures of NiFe_2O_4 (NFO)@ BiFeO_3 (BFO)@polypyrrole (PPy) benefiting from the dual properties of ferromagnetism and magnetostriction of NFO, the dual properties of antiferromagnetism and piezoelectricity of BFO, combined with the strong ability of electron polarization of the conductive polymer PPy.

Q9: *The authors must provide a detail discussion about the establishment of the dual-pinning effect resulting in optimized impedance and enhanced attenuation at low-frequency bands, leading to better EW performance.*

Reply: We thank the reviewers for the careful review. Firstly, the magnetic pinning effect in dual-pinning mechanism can achieve impedance matching at low-frequency bands. The combination of ferromagnetic NFO and antiferromagnetic BFO produces a magnetic pinning effect, which generates an exchange bias field and increased the magnetocrystalline anisotropy field. The natural resonance frequency is determined by the magnetocrystalline anisotropy field. Therefore, the natural resonance frequency can be increased to the GHz range by increasing the magnetocrystalline anisotropy field of magnetic components to obtain greater magnetic permeability. Theoretical analysis confirms that magnetic materials with special magnetic permeability values can effectively adjust the impedance and dissipation ability independently. Therefore, the magnetocrystalline anisotropy of materials can be effectively improved by the construction of magnetic pins. Then the low frequency permeability of the material is improved, and the impedance matching at low-frequency bands is optimized.

Moreover, dual-pinning mechanism provides enhanced electromagnetic wave loss. The higher permeability obtained by the magnetic pinning effect results in higher magnetic losses. Additionally, the combination of magnetostrictive NFO and multiferroic BFO produces a magnetoelectric driver, which can generate an internal bias electric field under the action of the exchange bias field generated by the magnetic pinning effect. The conductive polymer polypyrrole is employed for coating, which is rich in dipoles, and the generation of internal bias electric field plays a role of electric field pinning on the inversion of dipoles, strengthening the relaxation of heterostructure and dielectric loss. Leveraging the interface-induced dual-pinning effect, the magnetic pinning effect enhances the imaginary part of the permeability of the material at low frequency, leading to more adaptable impedance matching. Meanwhile, the dielectric

pinning improves the loss of low frequency medium and realizes excellent absorption at low-frequency bands.

To address this comment, corresponding description was added in the revised version.

Page 3, Paragraph 1, Line 3-8 in the revised manuscript:

The interfacial magnetic pinning effect of the antiferromagnetic BFO on the ferromagnetic NFO is achieved by the magnetic bias effect, which effectively improves the magnetocrystalline anisotropy of the material, which in turn improves the low-frequency permeability and optimizes the impedance in the low-frequency bands. Meanwhile, the internal bias electric field generated by magnetoelectric driver NFO@BFO plays a role of electric field pinning on the inversion of dipoles, strengthening the relaxation of heterostructure and dielectric loss.

Q10: *Figure 2 shows the schematic diagram of the magnetic pinning mechanism. b, M-H curves. c, Enlarged M-H curves for NFO@BFO and NFO/BFO. d, SFD plot. e, M vs 1/H² plots. Analysis of magnetic parameters of NFO@BFO@PPy and NFO/BFO@PPy f, permeability real part. g, permeability imaginary part. Authors must explain why they did not do the magnetic loops analysis of NFO@BFO@PPy and NFO/BFO@PPy composites.*

Reply: Thanks for the reviewer's constructive suggestion. Our primary objective is to investigate the intrinsic magnetic pinning effect, where the NFO@BFO core plays the role of magnetic pinning. To validate the magnetic pinning effect in static states, we chose to compare NFO@BFO with NFO/BFO (the influence of PPy on the magnetic properties is negligible), which is more conducive to elucidating the mechanism of electromagnetic wave absorption.

Taking the reviewer's suggestion into consideration, in view of the hysteresis loop studies under PPy presence conditions, we also conduct the hysteresis loops analysis of NFO@BFO@PPy and NFO/BFO@PPy, revealing analogous patterns to those of NFO@BFO vs. NFO/BFO, further supporting our proposed role of magnetic pinning. Relevant data has been discussed in Supporting Information.

Fig. R4 M-H curves for NFO@BFO@PPy and NFO/BFO@PPy.

Q11: Why NFO@BFO@PPy have a large magnetic permeability? Explain with more details.

Reply: Natural resonance can be considered as a special form of ferromagnetic resonance, located in the frequency interval 0.1 GHz-10 GHz, where an extreme value of the complex permeability occurs. The natural resonance frequency corresponds to the point at which the imaginary part of the permeability reaches maximum (*J. Magn. Mater.*, 2004, **281**, 195-205). For conventional spinel ferrite, the intrinsic resonance frequency is usually close to or below 1 GHz (*J. Magn. Mater.*, 2000, **215-216**, 253-259), and the natural resonance frequency is determined by the magnetocrystalline anisotropy field (*J. Magn. Mater.*, 2023, **586**, 171159). The permeability of NFO@BFO@PPy samples is mainly affected by the NFO@BFO magnetic core. There is magnetic pinning in the NFO@BFO core of the NFO@BFO@PPy sample. The enhancement of the NFO@BFO for magnetocrystal anisotropy can be illustrated by assuming that the FM magnetic moment of the NFO particles generates an exchange magnetic field $\mu_0 H_{ex}$, acting on the uncompensated magnetic moment of the interface AFM of the BFO shell layer. As a result, the frequency of the natural resonance increases and the imaginary part reaches a peak, and the NFO@BFO@PPy sample has a larger imaginary part of permeability. At the same time, NFO@BFO shows a larger saturation magnetization due to the magnetic pinning effect, which corresponds to a larger real part of the permeability. To sum up, NFO@BFO@PPy have a large magnetic permeability.

Q12: *The smaller coercivity voltages indicate that the strain generated in the magnetostrictive NFO core is effectively transferred to the piezoelectric material BFO shell, which promotes the polarization reversal process in the BFO.? Explain and justify.*

Reply: As in the polarization reversible process, the strain generated in the core of the magnetostrictive NFO is effectively transferred to the shell layer BFO, the BFO shell produces a surface polarization that promotes its polarization reversal thus obtaining a smaller recalcitrance field (*Nanoscale*, 2014, **6**, 8515). Piezoresponse force microscopy (PFM) was used to characterize the magnetoelectric conversion properties of core-shell NFO@BFO. This technique involves placing a conductive tip in contact with the surface of an NFO@BFO sample located on a conductive substrate, and an alternating voltage was applied to create a perpendicular electric field between the tip and the sample. The piezoelectric response of the sample was sensed by cantilever deflection. Localized piezoelectric response hysteresis loops are obtained at random positions of the nanoparticles by scanning the applied DC bias with and without an applied magnetic field and measuring both phase and amplitude responses. The phase loops obtained by PFM demonstrate that the BFO shells exhibit polarization reversibility with or without the presence of an applied magnetic field, and the polarization direction can be switched between the two polarities of the DC bias voltage at the tip. The asymmetric butterfly curves and D-E hysteresis loops are caused by the internal switched bias field. In the absence of magnetic field, the coercivity voltages of the BFO shell are -31.97 V and 55.81 V, while the coercivity voltages with the application of magnetic field are -23.99 V and 32.04 V. The smaller coercivity voltages indicate that the strain generated in the magnetostrictive NFO core is effectively transferred to the piezoelectric material BFO shell, which promotes the polarization reversal process in the BFO. So we demonstrate by PFM that under the action of the magnetic field, the NFO@BFO has a smaller coercive recalcitrance voltage, which can be shown to be a magnetostrictive field (*Adv. Mater.*, 2019, **31**, 1901378).

Q13: *A much lower RL_{min} of -65.30 dB can be achieved at 5.68 GHz for NFO@BFO@PPy with a thickness of 4.43 mm. (Fig. 4,a,b). The effective absorption bandwidth (EAB) of the NFO@BFO@PPy composite is 2.34 GHz (4.72 ~ 7.04 GHz) and shifts toward lower frequencies with a fill rate of only 15.0% (Supplementary Fig. 15). Why the thickness was too large. Explain and justify*

Reply: According to the 1/4 wavelength cancellation equation, the thickness corresponding to the peak of reflection loss of electromagnetic wave absorbing material inevitably increases with the decrease of frequency. Therefore, the thickness of the material is generally larger for the absorption of electromagnetic waves at lower frequencies. Compared with similar low-frequency absorbing materials, the thickness in our work is in the middle or upper level in related fields (*J. Mater. Chem. A*, 2021, **9**, 24571–24581).

$$t_m = \frac{n\lambda}{4} = \frac{nc}{4f\sqrt{\epsilon_r\mu_r}} \quad n = 1,3,5\dots$$

Where t_m refers to the thickness of the sample, λ presents the wavelength of the electromagnetic wave, f presents the frequency of the electromagnetic wave, c is the velocity of light in free space ($3 \times 10^8 \text{ m} \cdot \text{s}^{-1}$), ϵ_r is the complex permittivity, μ_r is the complex permeability.

Filler	RL_{min} (dB)	f_m (GHz) ^a	Thickness (mm)	Ref.
GO@TiO ₂ /TiO ₂ -C	-64.4	7.40	5.0	Carbon , 2021, 178 , 144–156
FeCo@C	-35.9	6.20	4.0	Adv. Funct. Mater. , 2023, 33 , 2213258
PMMA/PAN/ZIF-67	-57.1	3.90	4.6	ACS Appl. Mater. Interfaces , 2023, 15 , 31720–31728
CNT/Ni-MOF	-24.3	4.50	5.0	Nanotechnology , 2020, 31 , 394002
(Ni/C)/ZnFe ₂ O ₄	-52.6	5.81	4.5	J. Magn. Magn. Mater. , 2023, 568 , 170405
FeCo/Co@NC	-59.6	5.44	4.8	J. Colloid Interface Sci. , 2023, 650 , 1434–1445
P-CNF/Fe	-39.7	4.21	4.5	Ceram. Int. , 2019, 45 , 4474–4481
Graphene–Fe ₃ O ₄	-40.4	7.00	5.0	J. Appl. Phys. , 2013, 113 , 024314
γ -Fe ₂ O ₃ nanocubes/graphene	-57.2	5.12	5.3	Chem. Eng. J. , 2021, 405 , 126676
MnO ₂ @nanoporous carbon	-57.2	4.90	4.6	Chem. Eng. J. , 2021, 417 , 128087
NiCo ₂ O ₄ /Co ₃ O ₄ /NiO	-57.0	4.93	5.9	J. Mater. Chem. C , 2017, 5 , 3770–3778
Ni@C/G	-45.5	6.20	4.8	Chem. Eng. J. , 2020, 382 , 122980

NiFe ₂ O ₄ @BiFeO ₃ @PPy	-65.3	5.68	4.4	This Work
---	-------	------	-----	-----------

f_m : the frequency where RL_{min} is obtained.

Table R1 EWMA performance in low-frequency bands of typical lower-frequency absorbers.

Q14: Why the RCS reduction is too high in NFO@BFO@PPy?

Reply: The Radar Cross Section (RCS) is a physical quantity used to characterize the scattering of radar radiation by a target. It represents the proportion of reflected radar beam power received by the target in a given direction.

The simulation model we set up consists of a perfect electrical conductor (PEC) layer at the bottom and an absorbing layer with a thickness of 4.43 mm on top. The overall plan of the model is square with a length equal to the width of 200 mm. The sole variable in the simulation is the electromagnetic parameters of the absorber layer. Therefore, the RCS can be evaluated by the power reflection coefficient used to express the ratio of the power of the reflected electromagnetic wave to the power of the incident electromagnetic wave. Reflection loss (dB) is defined by (*J. Mater. Sci.*, 1998, **33**, 4971–4976):

$$RL (dB) = 10\log_{10}\left(\frac{U_{reflection}}{U_{incidence}}\right)$$

$$\frac{U_{reflection}}{U_{incidence}} = 10^{RL (dB)/10}$$

where $U_{reflection}$ represents the reflected electromagnetic wave power and $U_{incidence}$ represents the incident electromagnetic wave power, and the power reflection coefficient can be derived from the reflection loss. As shown in Fig. R3, the power reflection coefficient of NFO@BFO@PPy at 5.68 GHz with a thickness of 4.43 mm is lower than that of NFO/BFO@PPy. Therefore, NFO@BFO@PPy possesses a lower RCS.

Fig. R5 The power reflection coefficient of NFO@BFO@PPy and NFO/BFO@PPy. (at the thickness of 4.43 mm)

Q15: Why ethylene glycol and *p*-toluenesulfonic acid was used? Explain its importance.

Reply: Ethylene glycol plays an important role in the sol-gel synthesis of BiFeO₃ nanomaterials. Firstly, it can be used as a proficient solvent to dissolve Fe(NO₃)₃ and Bi(NO₃)₃ to form a homogeneous precursor solution and thus ensuring the formation of a homogeneous gel during the preparation by the sol-gel method, which is crucial for the coating of BiFeO₃ on the surface of NiFe₂O₄. Secondly, ethylene glycol can be used as a dehydrating agent, which is favorable for the formation of pure phase BiFeO₃ (*J. Mater. Chem. C*, 2016, **4**, 4092–4124; *J. Supercond. Nov. Magn.*, 2014, **27**, 1569–1577; *J. Magn. Magn. Mater.*, 2006, **304**, e772–e774; *Materials*, 2019, **12**, 1444).

To further verify the importance of ethylene glycol, similar organic solutions containing varying numbers of hydroxyl groups were selected for comparison (ethanol, ethylene glycol, 1,2-propanediol, 1,4-butanediol, and propanetriol). It can be observed that Bi(NO₃)₃ undergoes an alcoholysis reaction in ethanol and 1,4-butanediol, which produces a white precipitate and prevents the formation of a precursor solution, which is unfavorable for the homogeneous encapsulation of BiFeO₃. When using propanetriol, the high boiling point of propanetriol (290°C) makes it challenging to form a dry gel. 1,2-Propanediol, similar to ethylene glycol, is capable of dissolving Fe(NO₃)₃ and Bi(NO₃)₃ and forming a dry gel. However, XRD analysis revealed that the sintered product that the BiFeO₃ obtained using ethylene glycol contains the least impurities compared to other organic solutions. Thus, we chose ethylene glycol for the critical step of preparing BiFeO₃.

Fig. R6 a) Photographic representation of precursor solutions prepared using different organic solutions. (a1. Ethanol, a2. Ethylene glycol, a3. 1,2-propanediol, a4. 1,4-butanediol, a5. Propanetriol.). XRD patterns of BFO obtained using different solvent precursor solutions. b) Ethanol, c) Ethylene glycol, d) 1,2-propanediol, e) 1,4-butanediol, f) Propanetriol.

The addition of p-toluenesulfonic acid as a commonly used PPy dopant during PPy synthesis affects the precipitation of PPy π -electrons and alters its carrier mobility, which can improve the electrical conductivity of PPy, and thus enhance the dielectric constant of the material. Meanwhile, doping of p-toluenesulfonic acid in PPy increases conformational defects in the polymer structure, such as distortions and linkages (*J. Am. Chem. Soc.*, 2010, **132**, 1754–1755). In conclusion, as shown in Fig. R5, PPy doped with p-toluenesulfonic acid exhibit higher dielectric constants compared to undoped PPy, which can better improve the dielectric loss of the material and exhibit better electromagnetic wave absorption.

Fig. R7 The ϵ' (a), ϵ'' (b) and $\tan \delta\epsilon$ (c) of NFO@BFO@PPy (TsOH) and NFO@BFO@PPy (Non-TsOH). (d) The reflection loss of NFO@BFO@PPy (Non-TsOH)

Q16: Which version of COMSOL Multiphysics software is used. Mention in the text.

Reply: In this study, we performed simulations in the COMSOL Multiphysics 6.0 software. Related illustration has been added in the experimental section.

Page 12, Paragraph 5, Line 1-2 in the revised manuscript:

“In this study, we performed simulations in the COMSOL Multiphysics 6.0 software based on similar examples from literature.”

Q17: CST STUDIO SUITE was applied to simulate the RCS values of as-prepared NFO@BFO@PPy and NFO/BFO@PPy composites under open boundary conditions. Mention the version in the text.

Reply: CST STUDIO SUITE 2014 was applied to simulate the RCS values of as-prepared NFO@BFO@PPy and NFO/BFO@PPy composites under open boundary conditions. Related illustration has been added in the experimental section.

Page 15, Paragraph 5, Line 8-9 in the revised manuscript:

“CST STUDIO SUITE 2014 was applied to simulate the RCS values of as-prepared NFO@BFO@PPy and NFO/BFO@PPy composites under open boundary conditions.”

Q18: Conclusion must be brief

Reply: We have been succinct about the conclusion based on reviewers' requests. The

succinct conclusion is as follows:

“In summary, a novel low-frequency absorption mechanism based on a magnetoelectric bias interface-induced dual-pinning mechanism is proposed in this work and validated by constructing a bilayer core-shell structure NFO@BFO@PPy. The establishment of a dual-pinning mechanism based on the synergistic effect of magneto-electric coupling allows the optimization of low-frequency impedance matching and attenuation enhancement, thus improving the effective loss of low-frequency EM waves. EMWA tests and simulations further validate the superiority of this mechanism from both experiments and theories together, achieving a RL_{min} of -65.30 dB at a thickness of 4.43 mm, and an EAB that almost covers the c-band (4.72 ~ 7.04 GHz) with a fill rate of only 15.0%. This work not only broadens the way for the research of low-frequency EMWA materials, but also can support the improvement of the database of EMWA materials.”

REVIEWERS' COMMENTS

Reviewer #1 (Remarks to the Author):

The author has made good answers to the questions raised by the reviewers. This version is now ready to be accepted and published.

Reviewer #2 (Remarks to the Author):

The authors have incorporated suggested changes, I recommend the paper for publication.

RESPONSE TO REVIEWERS' COMMENTS

Reviewer 1

Reply:

Dear Reviewer,

We are very thankful to you for your valuable suggestions on our work and we are highly satisfied with your high rating. Based on your comments, we have revised and improved the article accordingly.

Thank you very much for your guidance on the structure and logic of the article. We have revised the links between paragraphs to make the structure of the article clearer and smoother. In addition, we have added more explanations of related concepts to help readers better understand the vein and content of the article.

In the experimental section, based on your suggestions, we have made detailed revisions and additions to the experimental data section so that readers can better understand and reproduce our study. We have added more details of the experimental data so that readers can more accurately understand our experimental results.

Your comments and suggestions are very important to our article. We will make changes according to your suggestions and make sure that our argument is clearer, more accurate and rigorous. Thank you for your support and help.

Finally, we would like to reiterate our thanks to you for your review and guidance. Your suggestions have been extremely helpful in improving the quality of our paper. If there are any other questions or need for further discussion, we always welcome your feedback, and we will reply and provide solutions as soon as possible!

Reviewer 2

Reply:

Dear Reviewer,

We appreciate that you have provided us with your review comments on our manuscript, and we are very satisfied with your high evaluation. Under your guidance, we have thoroughly revised and improved the manuscript.

We have added citations and references to relevant literature in the article based on your suggestions. We have researched more literature and cited studies from different perspectives to enhance the basis and credibility of our argument.

Your guidance on the structure and logic of the article is much appreciated. We have reorganized and optimized the paragraphs to improve the coherence and readability of the article.

More importantly, based on your suggestions, we have made detailed modifications and additions to the methods and experimental sections so that readers can better understand and reproduce our study. We added more experimental details, including information on experimental conditions and experimental procedures. We have also provided methods for analyzing relevant data and results so that readers can understand our experimental results more accurately. We believe that these modifications will enable readers to better understand the process and results of our study.

In the meantime, we greatly appreciate you pointing out the grammatical and spelling errors in the article. We have carefully reviewed and revised the article to ensure grammatical and spelling accuracy.

Many thanks again for the valuable comments and suggestions you have provided on our article. We are confident that this revision will result in a significant improvement in the quality and readability of the article. If you have any other comments or suggestions, please feel free to contact us at any time